# Relationship between Psychological Distress and Demographic Characteristics among Patients Undergoing Coronary Artery Bypass Graft (CABG) Surgery

**DOI:** 10.3390/healthcare10091763

**Published:** 2022-09-14

**Authors:** Noor Hanita Zaini, Khatijah Lim Abdullah, Raja Amin Raja Mokhtar, Karuthan Chinna, Shahrul Bahyah Kamaruzzaman

**Affiliations:** 1Department of Nursing Science, Faculty of Medicine, Universiti Malaya, Kuala Lumpur 50603, Malaysia; 2Department of Nursing, School of Medical and Life Sciences, Sunway University, Kuala Lumpur 47200, Malaysia; 3Department of Surgery, Faculty of Medicine, Universiti Malaya, Kuala Lumpur 50603, Malaysia; 4Faculty of Business and Management, UCSI University, Kuala Lumpur 50603, Malaysia; 5Department of Medicine, Faculty of Medicine, Universiti Malaya, Kuala Lumpur 50603, Malaysia

**Keywords:** coronary artery bypass graft, anxiety, depression, pain

## Abstract

Coronary artery bypass graft (CABG) surgery patients often present with anxiety and depression. These symptoms coupled with pain are major concerns and are widely reported among CABG patients. The study aimed to evaluate the relationship between psychological distress and patients’ demographic data. This cross-sectional study in which 178 patients from a surgery ward were selected using the census method. Data were collected using the Hospital Anxiety and Depression Scale and the Brief Pain Inventory (Short Form). The participants’ mean age was 57.49 ± 13.78 years. The majority of participants had a moderate level of anxiety (89.3%) and a moderate level of pain interference (74.7%). Significant differences were noted, with a higher proportion of moderate anxiety level participants aged between 60 and 74 years old, married, and with a tertiary level education (*p* < 0.05) being found. There were also significant differences between gender (*p* < 0.05) and pain severity (*p* < 0.01) across the levels of depression. The levels of anxiety, depression, and pain were significant, especially among older patients. These symptoms should be routinely assessed, and further identification of predictors such as socioeconomic status before the procedure is necessary.

## 1. Introduction

Coronary artery disease (CAD) is one of the leading causes of morbidity and mortality in both developing and developed countries [1]. According to World Health Organization data published in 2020, CAD death in Malaysia reached 36,729 or 21.86% of total deaths, which ranks Malaysia 61th in the world [2]. Although interventions such as percutaneous transluminal coronary angioplasty and stent implantation are widely used, many patients still require a Coronary artery bypass grafting (CABG) procedure. Coronary artery bypass grafting (CABG) is a surgical procedure to improve coronary circulation to adequately deliver oxygen and nutrients to the myocardium. After CABG surgery, the blood flow and oxygen supply to the heart are restored, and relief from angina or other CAD symptoms follows. The outcomes usually are excellent with 85% of people having significantly reduced symptoms, fewer future heart attacks, and a decreased chance of dying within 10 years following CABG surgery [3]. Thus far, it is the best treatment option for CAD. In Malaysia, the statistics from the National Heart Institute (NHI) show that the number of open-heart surgeries performed at the NHI increased gradually from 1000 cases in 2012 to 1200 cases in 2016 [4].

Patients scheduled for CABG and those recovering from the surgery may face considerable physical and psychological challenges. In this paper, psychological distress refers to symptoms of anxiety and depression levels alone. Other than psychological distress, pain is also the most common symptom patients seek for treatment. It has been reported by Woldegerima Berhe et al. [5] that among the older age (≥60 years) undergoing surgery, preoperative pain was found significantly associated with preoperative anxiety. Consistent with the existing literature, a study by NoorHanita et al. [6] stated that more than 50% of CABG patients in the Malaysia setting had moderate levels of anxiety, with scores ranging from 8 to 14 on the HADS-A, while 51.7% of the patients had mild depression. Depression and CAD are highly comorbid conditions, with the estimation of comorbidity ranging from 14 to 47% [6]. Some patients believe that having a CABG means the difference between life and death and that it carries a poor prognosis far worse than cancer. Pain is often distressing for patients if not adequately managed [7,8] and may lead to anxiety, depression, fatigue, a desire for death, poor quality of life, limitations in Activity of Daily Living (ADLs), poor compliance with treatment, and prolonged hospital stay [9,10]. Many factors influence the experience of anxiety, depression, and pain, which may be different for everyone. It seems that the concern with an altered mental image of the body was very important to these patients as with other concerns. Many patients contemplate death, recurrent heart attacks, and the possibility of leaving their loved ones while undergoing CABG [11]. Hence, a strong insight into the correlation between anxiety, depression, and pain can guide clinicians and other healthcare providers in the selection of both pharmacologic and non-pharmacologic measures for comprehensive pain control. Furthermore, during the study period, the challenges that nurses face during pre and postoperative management, including assessment of pain, language differences, and different pain perceptions among patients from different demographic backgrounds, make the nurses feel stressed and patient also feeling anxious. A study identified 150 nurses and looked at their demographics, confidence level, and identified stressors [12]. They found that the majority of these nurses identified challenges in the practice environment such as stress, deficits in clinical skills and knowledge, and feelings of inadequate preparation for their position. There is limited research on cardiac nurses’ pain knowledge, assessment practices, and postoperative pain management in the context of cardiac surgical patients.

In addressing this situation, nurses are required to assist patients in managing these symptoms during the CABG phase as high levels of anxiety and depression can cause distress to the patients and result in poor outcomes. Patients who are more anxious prior to the CABG procedure would encounter more postoperative pain, higher readmission rates, less long-term relief of cardiac signs and symptoms, and possibly death after discharge [13]. When the anxiety, depression, and pain levels are identified by nurses, they can address the issues affecting the patient by manipulating the environment and by providing appropriate care. There is ample evidence that adequate cardiac surgery education can reduce a patient’s level of anxiety, depression, and pain while also improving postoperative recovery [14,15,16,17]. Therefore, the study aimed to evaluate the relationship between psychological distress and patients’ demographic data among patients awaiting CABG surgery in the Malaysian hospital setting. The results of this study are expected to serve as a reference for nurses to pay increased attention to any psychological and physical changes, especially among patients who are more susceptible to anxiety and depression. Nurses also should try to meet specific nursing needs in preventing more serious complications such as mental disease.

## 2. Materials and Methods

### 2.1. Data Collection

This was a cross-sectional study that utilized structured questionnaires to assess the level of anxiety, depression, and pain among patients undergoing CABG surgery. The study was conducted in the surgical ward of a Tertiary Referral Centre based in Kuala Lumpur, Malaysia, between September 2014 and October 2015. The present report was prepared in keeping with the standards set forth by the Strengthening the Reporting of Observational Studies in Epidemiology (STROBE) statement [18].

The research population consisted of randomly selected patients scheduled for CABG surgery. Inclusion criteria were patients aged more than 18 years old, undergoing CABG surgery for the first time, who were alert and aware of time, place, and person, able to write, read and speak in both English and Malay languages; and have no known neurologic problems. Patients who were arranged to undergo emergency surgery and had clinical instability defined as decompensated heart failure or any acute process that causes major symptoms were excluded from this study. The sample size was calculated to be 180 according to the study conducted by Shahmansouri et.al., with prevalence of 39% moderate level of fear (p), 95% confident coefficient (z), and a detection accuracy of 0.07 (d) [11,19].

One hundred and eighty patients were randomly selected from the CABG waiting list. Those selected were approached to participate in this study; however, two patients refused to participate in the study, leaving a total of 178 CABG patients. Patients undergoing CABG surgery are usually admitted two days before the surgical procedure to allow for physical examination and laboratory studies. On the day before surgery, all participants were given a thorough explanation of the study process before signing consent forms. They also were asked if they were on any treatment for anxiety, depression, or chronic pain. They were expected to complete all of the questionnaires, which took approximately 15 to 20 min. Some patients had to be assisted with the completion of the questionnaire for various reasons including poor eyesight.

### 2.2. Instruments

Three data collection tools were used in the study: The ‘Personal Information Form’ was used to collect patients’ characteristics, the ‘Hospital Anxiety and Depression Scale (HADS) was used to determine the level of anxiety and depression [20], and the Brief Pain Inventory (Short Form) (BPI-SF) was used to determine the level of pain severity and pain interference among patients who underwent CABG surgery [21].

For the ‘Personal Information Form’, the demographic data, which includes age; gender; marital status; educational level, and monthly income were collected through individual interviews with participants; while medical comorbidities, which include hypertension, diabetes mellitus, history of myocardial infarction, and others (such as Musculoskeletal disease, Bowel disease, or Obesity) and any treatment for chronic pain before CABG were collected from medical records and patients. The researcher also administered mini-mental state examinations to patients over the age of 65 to ensure that the participants were alert and orientated.

The HADS and BPI-SF have been repeatedly validated and demonstrated to be effective screening tools in assessing the presence and severity of both symptoms [22,23]. The HADS is a brief, user-friendly, self-report questionnaire developed by Zigmond and Snaith in 1983 to assess the levels of anxiety and depression among patients in nonpsychiatric hospitals. Thus, the researcher used the HADS to measure anxiety and depression in hospitalized patients awaiting CABG surgery. This questionnaire comprises 14 items in two subscales: seven for anxiety (HADS-A) and seven for depression (HADS-D). Each item is rated on a 4-point Likert scale from 0 (not at all) to 3 (often), resulting in a potential range of scores from 0 to 21 for each subscale. Scores of 0 to 7 indicate mild anxiety or depression; 8 to 14 indicate moderate anxiety or depression, and 15 to 21 indicate a severe form of anxiety and depression [20]. It has been reported that both the English and Bahasa Malaysia versions of the HADS have a sensitivity and specificity of approximately 0.80 [23]. To ensure the quality and relevance of the questions among CABG patients, face validity was assessed by an expert panel consisting of the Head of the Department of Nursing Science, a Clinical Specialist Cardiothoracic Surgeon, a ward nurse manager of the Cardiothoracic Intensive Care Unit, and a Clinical Specialist Cardiac Rehabilitation. The validity and reliability of the questionnaire have been assessed in previous studies [24,25]. In this study, the internal consistency reliability of Cronbach’s alpha for HADS among CABG patients was 0.74 for anxiety scores and 0.67 for depression scores.

Perceived pain was measured with the Brief Pain Inventory (Short Form) (BPI-SF). It is a well-established eleven-item questionnaire that consists of a pain severity subscale to rate pain severity in four domains (worst, least, average, and right now) and a pain interference subscale to measure the perceived degree of pain that interferes with daily activities in seven functional domains (general activity, mood, walking ability, work, relations with others, sleep, and enjoyment of life). Each item is rated on a visual analog scale of 10 cm, where 0 indicates ‘none’ and 10 indicates ‘worst imaginable’. In this study, some of the patients verbalized that the pain was due to chest discomfort, body ache, bowel problems, and musculoskeletal disease. Patients were asked to rate their pain by circling one number that described how the pain has interfered with their life such as their sleep, for over the previous 24 h. The seven BPI interference items were summed, thus generating scores ranging from 0 to 70. The cut points (CP) for BPI among CABG patients are 1–3 for mild pain, 4–6 for moderate pain, and 7–10 for severe pain [26]. Cronbach’s alpha for the pain-severity and pain-interference scores in this study was 0.72 and 0.80, respectively. The BPI has been translated into a number of languages with validation studies.

Thirty (30) patients undergoing CABG surgery who have the same characteristics as the study sample were invited for the pilot study to check the reliability and suitability of the instrument in the Malaysian setting, using Cronbach’s alpha coefficient. In this study, the internal consistency reliability of Cronbach’s alpha for HADS among CABG patients was 0.74 for anxiety scores and 0.67 for depression. Cronbach’s alpha for the pain-severity and pain-interference score was 0.72 and 0.80, respectively. All the survey sections remain unchanged as the reliability had been demonstrated and face validity established.

### 2.3. Ethical Consideration

This study was conducted in accordance with the Declaration of Helsinki and the Caldicott Principle. Meanwhile, ethical approval was obtained from the Medical Research Ethics Committee (MREC) of University Malaya Medical Centre, Kuala Lumpur, Malaysia (MEC ID No. is: 201401-0709), where the study was conducted. Patients received written and verbal information regarding the flow and aims of the study as well as the potential benefit of consenting and participating in the study. Patients were informed that they could withdraw from the study at any time without the need to disclose their reason and that there would be no negative consequences to their medical care. The researcher emphasized that participation in the study was entirely voluntary and that the subject’s confidentiality and anonymity will be assured through the coding of all data. Finally, the safety of the soft and hard copy data was ensured.

### 2.4. Data Analysis

All statistical analyses were carried out via SPSS Software Version 23.0 (IBM Corp., Armonk, NY, USA). The frequencies and significance of psychological symptoms and pain levels were individually calculated through the Chi-square test for dichotomous variables with a *p*-value less than 0.05 indicating statistical significance. The estimation of each predictor (odd ratios and 95% confidence intervals) was calculated. Quantitative variables were expressed as mean ± standard deviation.

## 3. Results

### 3.1. Socio-Demographic Characteristics

The socio-demographic characteristics of the respondents are shown in Table 1. Data were collected from 178 patients: 127 males, and 51 females. The mean ± SD age of the sample was 57.49 ± 13.78, and ages ranged between 18 and 89 years old. They were primarily men (n = 127, 71.3%), married (n = 127, 71.3%), with a secondary education (n = 113, 63.5%), and Chinese (n = 119, 66.8%). Comorbidities were common, particularly a previous myocardial infarction (n = 60, 33.7%). With regards to the levels of anxiety and depression, the majority of participants had moderate anxiety levels (n = 159, 89.3%) and mild depression levels (n = 107, 60.1%) before CABG surgery. In terms of the level of pain, almost half of the study participants (n = 87, 48.9%) reported that they experienced mild pain severity and more than half (n = 133, 74.7%) experienced moderate pain interferences.

### 3.2. Association between Socio-Demographic Characteristics with Level of Pain and Psychological Distress

Table 2 shows the association between socio-demographic characteristics with the level of pain and psychological distress. There were no major differences in the anxiety level by gender, race, comorbid conditions, and level of pain. There were, however, socio-demographic characteristics that significantly differed across anxiety levels (*p* < 0.05), with a higher proportion of moderate anxiety level participants (89.3%) being aged between 60 and 74 years old (42.1%), married (73.6%), and those with secondary educational level (64.2%). There were also no remarkable differences in the proportion of the depression level by age group, marital status, education level, medical comorbid conditions, and pain interferences. However, gender (*p* < 0.05) and pain severity (*p* < 0.01) differed significantly across the level of depression. A higher proportion of mild depression was found in the male patients (76.6%) and among those who experienced mild pain severity (58.9%).

## 4. Discussion

The present study generally showed that a high number of patients undergoing CABG had a moderate level of anxiety, with a HAD-A score ranging from 8 to 14 (89.3%) and mild depression (60%). This is due to patients who are worried and have fear about surgery, chest wound, pain, mobility, and risk of death. Other than that, the increasing tendency towards early discharge from hospitalization and hospital costs can result in increasing psychological symptoms and nursing needs, especially during the homestay period. This finding is consistent with a previous study that showed a higher proportion of patients undergoing CABG surgery experienced increased anxiety because they were thinking about pain and risk of death [11,27].

In this study, the association between socio-demographic factors such as age, marital status, and education was found to be significantly associated with anxiety. The patients whose age was 60 to 74 were found to be more anxious compared to the patients whose age was between 15 and 29 years (*p* = 0.017). Age is significantly related to anxiety level, but not depression level. According to Weiss et al. [28], age is a major predictor of anxiety scores with increased age contributing to increased anxiety, whereas age was not a significant predictor of depression scores. This finding contrasted with a study by Nigussie et al. [29] that found a negative relationship between age and anxiety before CABG surgery, as younger patients are more anxious than elderly patients. It was also reported by Parvan et al. [30], who stated that low levels of anxiety can be expected in the majority of patients who were over 60, as patients in this age group typically have a history of heart disease and hospitalization, consequently making them more familiar with the situation that they were in. This was not so in our study, where 42% and 45% of those aged 60 to 74 years old had moderate levels of anxiety and depression, respectively, compared to the younger age group. This finding may reflect on various unmet needs among our older patients facing surgical interventions among other comorbid conditions in hospital, such as lack of access or understanding of current conditions or the details of pre and postoperative care and perhaps also concerns regarding the financial cost of the procedure and care.

Anxiety may be exacerbated by a lack of health literacy or lower educational status, both of which result in unfamiliarity with the healthcare environment, and those who were anxious are more likely to be unable to follow the postprocedural plan of care, resulting in adverse events [31,32]. In Malaysia, a lack of autonomy among older people regarding their health may also be related to sociocultural aspects where decisions on health are left to their offspring. Thus, the Western understanding of autonomy may not be wholly applicable in the Asian setting, especially in the setting of giving consent, determining best interests, and deciding on end-of-life care [33]. Our data revealed that marital status is also one of the factors associated with increased anxiety levels prior to CABG. There are some possible reasons related to marital status. Information about the surgery may induce thoughts of one’s own potential death, since death is mentioned as a possible complication of CABG surgery. Another possible explanation is that during their lives single status patients were more frequently confronted with thoughts of their own death.

There were also significant associations with gender and pain severity (*p* < 0.05) in our study patients. Based on the level of depression, the results of this study showed that pain severity and gender were significant predictors among CABG patients. A study by Hweidi, Gharaibeh, Al-Obeisat, and Al-Smadi, [34] stated that higher levels of depression were reported among female CABG patients as compared to male patients. The findings contradict previous research that suggests that female patients have higher anxiety levels than male patients [31,32]. However, male patients in this study showed a significantly higher level of depression compared to women (*p* < 0.05). Some authors have suggested that biological factors such as the influence of female sex hormones and psychological factors including women’s multiple roles as a result of recent societal changes could contribute to these differences between men and women [30,31]. Nurses need to pay increased attention to any emotional changes, especially among patients who are more susceptible to anxiety and depression. Based on these results, we may conclude that identification of the psychological distress and pain levels of patients undergoing CABG is critical for nurses because it allows them to prioritize and develop appropriate and effective interventions. In our study, the anxiety, depression, and pain that occurred before surgery could be addressed readily. The findings in this study have important implications for the practice of further assessments of pre-operative anxiety, depression, and pain in certain groups of patients as it may impact the course and prognosis of their recovery from surgery. Addressing anxiety, depression, and pain particularly for women living alone may be important targets for interventions to improve outcomes following cardiac surgery.

A study has shown that patients who received information regarding their surgery from their primary physicians reported greater depression levels than those who received the information from nurses [14,17,31]. This could suggest patients’ satisfaction with nurses who tends to be more approachable and responsive to patients’ learning needs. Thus, nurses play a crucial role in performing anxiety, depression, and pain levels assessments before and after surgery and their involvement should be investigated further. In our study population, patients did not receive any pre-operative formal education regarding the surgical process and what to expect following the procedure. The higher proportion of patients in our study that reported having psychological distress and a reduced ability to manage pain preoperatively, suggests that there is a need for intervention via education and support by the admitting team. This study has highlighted that there is a correlation between hospital anxiety and depression levels and the pain level among CABG patients. Prior to CABG surgery, the majority of patients had moderate anxiety and mild depression level. While in terms of pain interference, it was found to be positively correlated with a higher level of anxiety.

We suggest that a quick evaluation of anxiety, depression symptoms, and pain as a part of the preoperative workup could enable the identification of high-risk patients for whom pharmacological or nonpharmacological intervention could be instituted. This can be delivered by clinical nurse educators prior to and on admission. Furthermore, many hospitals offer stress management services and these resources could be used for appropriately identified cardiac surgery inpatients. To identify who is most at risk, there are studies that report that age, gender, and marital status can all have an impact on pre-and postoperative anxiety [14,29,33].

## 5. Strengths and Limitations

Understanding the associations between pain and psychological distress as well as contributing socio-demographic and medical factors will help to determine which patients are at a higher risk of adverse outcomes. Routine assessments should be performed before surgery to determine patients who are at risk for increased anxiety, depression, and pain after surgery. Nurses should be well trained to evaluate patients’ symptoms of psychological and physical problems. Recognizing all of these symptoms in patients awaiting surgery aids in the planning of appropriate interventions to improve such symptoms that are based on patients’ individual needs or even on health education groups, which helps in preparing the patients for surgical procedures. Further studies should be conducted focusing on the anxiety, depression, or pain of the cardiac patient that is related to the self-management or comparison between the group with and without intervention. The effective result will be determined in further research to improve the quality of life for cardiac patients.

Although this study provides several implications for nursing practice, there were some limitations in this study. The cross-sectional nature of this study restricts us from making causal inferences. A longitudinal study could aid in the investigations of the variations in the course of anxiety, depression, and pain associated with CABG surgery. Furthermore, the sources of anxiety were not investigated in this study, limiting the justifications of the results generated. Self-reported measures may include information bias due to the patient’s accurate recall or social desirability. However, investigations based on self-reporting have been proven to be reliable. Another drawback of this study is that the patients may have felt uncomfortable answering the questionnaire prior to CABG surgery, which puts them under additional pressure and stress. They may have given different responses if the questionnaires were completed after the surgery. Because of the limited time available to answer the questionnaire, there may be bias in reporting symptoms. Moreover, convenience sampling and the use of a single tertiary cardiac center can limit generalizability and lead to selection bias. In particular, to reach more accurate conclusions, a larger geographical area taking these predictors into account should be studied.

## 6. Conclusions

CABG has become a relatively common surgical operation in recent decades. Most of the participants in this study had increased anxiety and depression as well as pain interference levels before surgery. The present study showed that the level of pain is associated with symptoms of depression in patients undergoing CABG surgery. The findings suggest the need for pre and postoperative assessments of psychological symptoms and pain among CABG surgical patients to enhance the patients’ abilities to perform self-care, which in turn can prevent complications. Therefore, intervention is warranted for patients before going through the CABG procedure. Intervening before surgery is crucial since patients who were anxious before the surgery are more likely to have continuous symptoms of anxiety throughout their recovery phase. This requires timely medical interactions and well-thought-out educational programs by nursing professionals to improve treatment outcomes and quality of life. Further studies are required to probe into individual life situations and anxiety of cardiac surgery patients if more effective educational programs are to be devised.

## Figures and Tables

**Table 1 healthcare-10-01763-t001:** Demographic characteristics with the level of anxiety, depression, and pain of the participants (N = 178).

Variables	No. of Participants n (%)
Age (mean ± SD)	57.49 (13.78)
Minimum	32
Maximum	89
Gender	
Male	127 (71.3)
Female	51 (28.7)
Marital Status	
Single	20 (11.2)
Married	127 (71.3)
Divorce/Widow	31 (17.4)
Educational Level	
None	7 (3.9)
Primary	22 (12.4)
Secondary	113 (63.5)
Tertiary	36 (20.2)
Monthly Income	
Less < USD 447.68	16 (8.99)
USD 447.90–671.52	86 (48.3)
USD 671.74–895.36	42 (23.6)
Above > USD 895.58	34 (19.1)
Comorbidities	
Hypertension	57 (32)
Diabetes Mellitus	22 (12.4)
Previous Myocardial Infarction	60 (33.7)
Others (Musculoskeletal disease, Bowel disease, Obesity)	39 (21.9)
Anxiety Level	
Mild	18 (10.1)
Moderate	159 (89.3)
Severe	1 (0.6)
Depression Level	
Mild	107 (60)
Moderate	67 (37.6)
Severe	4 (2.2)
Pain Severity	
Mild	87 (48.9)
Moderate	78 (43.8)
Severe	13 (7.3)
Pain Interference	
Mild	39 (21.9)
Moderate	133 (74.7)
Severe	6 (3.4)

**Table 2 healthcare-10-01763-t002:** Association between demographic characteristics and psychological distress (anxiety and depression) (N = 178).

		Psychological Distress							
		Level of Anxiety				Level of Depression			
	N (%)	Mild (0–7) (n = 18)	Moderate (8–14) (n = 159)	Severe (15–21) (n = 1)	*p*-Value	Mild (0–7) (n = 107)	Moderate (8–14) (n = 67)	Severe (15–21) (n = 4)	*p*-Value
Demographic Characteristics									
Age									
15–29	9 (5.1)	2 (11.1)	7 (4.4)	-	0.017 *	7 (6.5)	2 (3.0)	-	0.863
30–44	16 (9.0)	6 (33.3)	10 (6.3)	-		8 (7.5)	8 (11.9)	-	
45–59	62 (34.8)	3 (16.6)	59 (37.1)	-		39 (36.4)	21 (31.3)	2 (50.0)	
60–74	74 (41.6)	6 (33.3)	67 (42.1)	1 (100)		42 (39.3)	30 (44.8)	2 (50.0)	
75–89	17 (9.6)	1 (5.5)	16 (10.1)	-		11 (10.3)	6 (9.0)	-	
Gender									
Male	127 (71.3)	13 (72.2)	113 (71.1)	1 (100)	0.813	82 (76.6)	41 (61.2)	4 (100)	0.040 *
Female	51 (28.7)	5 (27.8)	46 (28.9)			25 (23.4)	26 (38.8)	-	
Marital Status									
Single	20 (11.2)	5 (27.8)	15 (9.4)	-	0.035 *	15 (14.0)	5 (7.5)	-	0.160
Married	127 (71.3)	10 (55.5)	117 (73.6)	-		79 (73.8)	45 (67.2)	3 (75.0)	
Divorced	31 (17.4)	3 (16.7)	27 (17.0)	1 (100)		13 (12.1)	17 (25.4)	1 (25.0)	
Education Level									
None	7 (3.9)	3 (16.7)	4 (2.5)	-	0.043 *	5 (4.7)	2 (3.0)	-	0.737
Primary	22 (12.4)	2 (11.1)	20 (12.6)	-		12 (11.2)	10 (14.9)		
Secondary	113 (63.5)	11 (61.1)	102 (64.2)	-		70 (65.4)	41 (61.2)	2 (50.0)	
Tertiary	36 (20.2)	2 (11.1)	33 (20.8)	1 (100)		20 (18.7)	14 (20.9)	2 (50.0)	
Monthly Income									
Less < USD 447. 0.68	16 (8.99)	6 (33.3)	10 (6.29)	-	0.215	2 (1.87)	12 (17.9)	2 (50.0)	0.060
USD 447.90–671.52	86 (48.31)	7 (38.9)	78 (49.1)	1 (100)		55 (51.4)	29 (43.3)	2 (50.0)	
USD 671.74–895.36	42 (23.6)	2 (11.1)	40 (25.2)	-		25 (23.4)	17 (25.4)	-	
Above > USD 895.58	34 (19.1)	3 (16.7)	35 (22)	-		25 (23.4)	9 (13.3)	-	
Medical Co-morbidities									
Hypertension	57 (32.0)	5 (8.8)	51 (89.5)	1 (1.7)	0.962	38 (66.7)	15 (26.3)	4 (100)	0.058
Diabetes Mellitus	22 (12.4)	2 (9.1)	20 (90.9)	-		13 (59.1)	9 (40.9)	-	
Previous Myocardial Infarction	60 (33.7)	12 (20)	48(80)	-		38 (63.3)	22 (36.7)	-	
Others (Musculoskeletal disease, Bowel disease, Obesity)	39 (21.9)	4 (10.3)	35 (89.7)	-		23 (59.0)	16 (41.0)	-	
Level of Pain									
Pain Severity									
Mild	87 (48.9)	9 (50.0)	77 (48.4)	1 (100)	0.410	63 (58.9)	22 (32.8)	2 (50.0)	0.008 *
Moderate	78 (43.8)	6 (33.3)	72 (45.3)	-		37 (34.6)	40 (59.7)	1 (25.0)	
Severe	13 (7.3)	3 (16.7)	10 (6.3)	-		7 (6.5)	5 (7.5)	1 (25.0)	
Pain Interference									
Mild	39 (21.9)	1 (5.6)	38 (23.9)	-	0.455	24 (22.4)	14 (20.9)	1 (25.0)	0.193
Moderate	133 (74.7)	16 (88.9)	116 (73.0)	1 (100)		80 (74.8)	51 (76.1)	2 (50.0)	
Severe	6 (3.4)	1 (5.6)	5 (3.1)	-		3 (2.8)	2 (3.0)	1 (25.0)	

** Chi-square test for level of anxiety and level of depression * Indicates a significant difference at *p* value < 0.05 * USD 1 = MYR 4.47.

## Data Availability

Data are available on request from the authors.

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
