# Peer review of "Relationship between Psychological Distress and Demographic Characteristics among Patients Undergoing Coronary Artery Bypass Graft (CABG) Surgery"

_healthcare, 2022, doi:10.3390/healthcare10091763_

Round 1

Reviewer 1 Report

Zaini, et al. investigated the levels of anxiety, depression, and pain in hospitalized patients scheduled to undergo CABG in Malaysia.  The study focuses on an important topic and is well-written.  However, I have several suggestions to improve the manuscript:

Comments:

·         -  The authors never evaluated pain level as an outcome in the study and rather focused on pain as a potential predictor of anxiety and depression.  The authors should either include analyses evaluating pain as an endpoint or remove “pain level” from the title and statement of study purpose.

·         -  With regards to selection of patients for the study, were patients screened for a history of anxiety, depression, or chronic pain?  This is an important consideration given the well-described link between these co-morbidities.

·         -  It appears that pain level and interference were reported prior to surgery.  Is this correct?  If so, this should be clearly stated so the reader does not interpret the findings as being applicable to post-surgical pain or persistent pain after CABG.  In addition, the causes of pre-surgical pain should be included as medical co-morbidities in Table 1.

·          - What methods were used to control for multiple comparisons in statistical testing? 

·         -  There are some inconsistencies in the numbers included in the Tables.  For example, Table 1 indicates that the minimum patient age was 32 and the maximum was 74.  However, the analysis reported in Table 2 indicates that 9 patients were age 15-29 and 17 patients were age 75-89.  In addition, the percentages reported for each of the covariates investigated appear to be based on the number of patients at that level of anxiety or depression.  However, this is not the case for the percentages reported for the medical co-morbidities, which seem to be based on the number of patients with that comorbidity.  Please review all tables and correct as necessary.

·       -   Some of the results reported in Tables 2 and 3 are difficult to interpret and appear to be conflicting.  For example, Table 2 indicates that marital status is associated with anxiety, but this relationship is not seen in Table 3, where single marital status is instead significantly associated with depression.  Monthly income is not significantly associated with anxiety or depression in Table 2, but income of RM2000-RM3000 is associated with significantly lower anxiety in Table 3.  Please clarify.

Author Response

Dear Reviewer;

Thank you

Reviewer 2 Report

In general, the authors have performed an interesting study about an important psychological aspect (anxiety and depression) of patients who undergo CABG and its impact on pain perception. Nevertheless, after reading the manuscript I have some questions and suggestions for the authors: 

1. I believe that the scope of the study is not clear and this is reflected by several parts of the manuscript. The authors should state clearly that they aimed at the evaluation of preoperative pain along with anxiety and depression as well as their associations.

2. You should improve the title of the manuscript in order to be more indicative of the study scope (i.e. Relationship between psychological distress and preoperative pain perception....). Also, in the abstract you should state that the aim of your study was preoperative evaluation of these variables in patients undergoing CABD surgery. 

3. Please, have a second look of the manuscript in order to improve some grammatic/syntax errors. Also, use formal expressions and brief sentences. 

4. Please, place the references appropriately in each sentence that you use. 

5. In the introduction, please place the scope of your study in a separate paragraph at the end. 

6. Also, in the first paragraph of introduction you could use more specific epidemiological data about CAD and CABG (i.e. incidence rates etc.). Similarly, you could do so for anxiety, depression and pain rates in patients who undergo CABG surgery.

7. In the introduction, you could merge 2nd and 3rd paragraph and provide more focused presentation about anxiety, depression, pain perception prior to CABG and their associations. Then, provide a separate paragraph about the challenges that nurses face regarding pre and postoperative pain management of these patients.

8. In the methods, please explain which comorbidities you have collected in your data.

9. In the data analysis section, please explain if you have checked the variables for normal distribution.

10. Also, since the scope of your study was to assess anxiety, depression and the effects on pain perception as well as their associations, you should use a regression analysis and specify which parameters you will include in this analysis (i.e. regression analysis about the parameters that affect preoperative pain including comorbidities).

11. Please, include in table 1 the results about the comorbidities of your study population (i.e. hypertension, diabetes mellitus,...).

12. In the results, you should provide a new table about the regression analysis describing the parameters that affect preoperative pain management. Please see comment 10 for explanation. In my opinion, table 3 is confusing and should not be used.

13. I think that discussion should be restricted in length and completely rearranged. The authors should present the major results in the 1st paragraph (including the regression analysis about preoperative pain perception). Then, in separate paragraphs discuss every result of their study in relation to the evidence from the literature in a comprehensive manner (i.e. a paragraph about socio-demographic parameters, another about anxiety, another about depression, another about pain perception). 

Author Response

Dear Reviewer

Thank you

Reviewer 3 Report

The paper is interesting, but the Introduction appears too long and repetitive and should be shortened. 

The fact that CABG could induce psychological distress and pain is already known. However, to frame this information into perspective the Authors should report in detail their contemporary information strategy and content for the patients undergoing CABG at their Institution. This could be the important background for evaluating the reporting of distress and pain by their patients.  Furthermore this is a descriptive paper of data related to 2014-2015 (why those data were not published before?) without any form of intervention strategy to compare their 2014-2015 approach to patients, with some new more structured and informative approach of 2022.

It shoudl be explained why the patients were randomly selected and not consecutive (page 3). How randomization procedure was performed ? Could this procedure and this type of inclusion have introduced a selection bias ? 

It is not clear why 30 patients were invited for a pilot study (page 4) and how they were managed in comparison with the total group? Were they then included in the total number of patients?

From the statistical point of view (page 4) why ANOVA analysis was not adopted to compare between and within group values ?

At page 4, Results; 3.1 it is reported that 124 patienst had CABG for the first time. Don't the Authors think that this can introduce an other heterogeneity bias into this study. 

In the last 3 lines of page 4 the results concerning pain experience are reported. It is not clear and it should be explained what kind of pain are referring to (psychological, physical, anginal, chest pain ?) since these patients are studied before CABG. 

In the Tables 1, 2 and 3 income values should be reported also as American Dollars equivalent to allow a better understanding of this information.  Abbreviations' meaning should be reported in the Legend of the Tables.

At page 12 , first line ..."higher number" .... comparing to what ?

At page 12, second paragraph:  It is said that "In our study population, patients did not receive a pre-operative formal education regarding the surgical process and what to expect following the procedure. " This it is not clear and it should be explained how the patients were informed about the fact that they have to undergo CABG and why. This is of great importance since only 20.2% of the patients are reported to have tertiary level of education (Table 2) and this type of information procedure could have largely conditioned the results. Was some additional information given about CABG after this distress and pain assessment?

Discussion appears also too long and repetitive and should be shortened. 

Author Response

Dear Reviewer

Thank you

Round 2

Reviewer 3 Report

The Authors answered to the questions.